# Validating the Transformation of PROMIS-GH to EQ-5D in Adult Spine Patients

**DOI:** 10.3390/jcm8101506

**Published:** 2019-09-20

**Authors:** Shreyas Panchagnula, Xin Sun, Julio D. Montejo, Aria Nouri, Luis Kolb, Justin Virojanapa, Joaquin Q. Camara-Quintana, Samuel Sommaruga, Kishan Patel, Nikita Lakomkin, Khalid Abbed, Joseph S. Cheng

**Affiliations:** 1Department of Neurosurgery, Yale School of Medicine, New Haven, CT 06511, USA; 2Dartmouth Hitchcock Medical Center, Lebanon, NH 03756, USA; 3Department of Neurosurgery, University of Cincinnati, Cincinnati, OH 45267, USA; 4Department of Neurosurgery, Geneva University Hospitals, 1205 Geneva, Switzerland; 5Donald and Barbara Zucker School of Medicine at Hofstra/Northwell, Hempstead, NY 11549, USA; 6Icahn School of Medicine at Mount Sinai, New York, NY 10029, USA

**Keywords:** EQ-5D, PROMIS, spine, transformation, quality of life, patient outcomes, validation

## Abstract

Spinal disorders and associated interventions are costly in the United States, putting them in the limelight of economic analyses. The Patient-Reported Outcomes Measurement Information System Global Health Survey (PROMIS-GHS) requires mapping to other surveys for economic investigation. Previous studies have proposed transformations of PROMIS-GHS to EuroQol 5-Dimension (EQ-5D) health index scores. These models require validation in adult spine patients. In our study, PROMIS-GHS and EQ-5D were randomly administered to 121 adult spine patients. The actual health index scores were calculated from the EQ-5D instrument and estimated scores were calculated from the PROMIS-GHS responses with six models. Goodness-of-fit for each model was determined using the coefficient of determination (*R*^2^), mean squared error (MSE), and mean absolute error (MAE). Among the models, the model treating the eight PROMIS-GHS items as categorical variables (CAT_Reg_) was the optimal model with the highest *R*^2^ (0.59) and lowest MSE (0.02) and MAE (0.11) in our spine sample population. Subgroup analysis showed good predictions of the mean EQ-5D by gender, age groups, education levels, etc. The transformation from PROMIS-GHS to EQ-5D had a high accuracy of mean estimate on a group level, but not at the individual level.

## 1. Introduction

High costs associated with surgical treatment of spine disorders demand a larger role for cost-utility analyses of treatment options. Amidst socioeconomic limitations and finite resources, spinal disorders occur at a high frequency, incur high costs for the healthcare system, and are treated with a heterogeneity of interventions. According to the 2010 Global Burden of Disease Study, low back pain had the greatest number of years lost to disability out of 291 conditions studied [1,2] and the annual direct costs of care provided for patients with spine disorders has been estimated at $90 billion [3]. Low back pain in particular presents a unique challenge, as there are numerous treatment modalities available whose comparative efficacy and value have not been fully substantiated [2].

Measuring the value of an intervention necessitates the use of a health utility score that encapsulates the health status, or patient-perceived overall health, at any given moment. Health status measures (HSMs) generally fall into two categories: (1) profile-based measures, such as the Patient-Reported Outcomes Measurement Information System (PROMIS) [4]; and (2) preference-based measures, such as EuroQol 5-Dimension (EQ-5D) [5]. Profile-based measures characterize health status by assigning a score to each of multiple domains of health. Preference-based measures characterize health status by providing a single utility score from multiple domains of health. The utility score, based on valuations of different health states, is central to estimation of quality-adjusted life years (QALY), cost-utility analysis, cost-effectiveness of interventions, and quantitation of health outcomes [2,6].

Many health status measures have been designed for generic or disease-specific use [7,8]. In 1990, EuroQol developed the EQ-5D three-level survey (EQ-5D-3L, abbreviated as EQ-5D below), a preference-based HSM with two parts: (1) a descriptive survey with five questions assessing five dimensions of health; and (2) a visual analog scale that permits a numeric self-assessment of general health [5]. Responses to the descriptive survey yields a health utility index score.

In 2007, the National Institute of Health (NIH) developed PROMIS Global Health Survey (PROMIS-GHS), a standardized, self-reported profile-based HSM with 10 self-reported global health items that summarize general perceptions of health [4]. This survey is freely available for public use and is increasingly adopted in clinical settings. However, economic analyses have been classically performed using other preference-based measures, including EQ-5D.

With an increased desire to determine the value of health care and increase in HSMs, there is a growing interest to correlate different HSMs. In 2009, Revicki et al. facilitated a conversion from PROMIS-GHS to EQ-5D index scores using generic United States (US) population data [9]. Since then, many clinical studies have used this model (REV_Reg_) when evaluating health outcomes of surgical and medical interventions [10,11,12]. While effective, such a conversion faces challenges and requires validation for specific patient populations or diseases. Furthermore, design of the model itself and its parameters can be optimized.

For instance, in 2017, Thompson et al. proposed new models to optimize REV_Reg_ using linear and equipercentile equating [13]. Linear and equipercentile equating are linking techniques that, after predicting scores, assign profile-based responses to preference-based scores by aligning score distributions of the two scales. Using Revicki et al.’s original data set, they recreated Revicki et al.’s regression model (REV_Reg_), applied linear equating to REV_Reg_ (REV_LE_), and applied equipercentile equating to REV_Reg_ (REV_equip_). In a similar fashion, they created three models by treating the score as categorical variables (CAT_Reg_, CAT_LE_, CAT_equip_) for a total of six models. They performed external validation of these models on a neurologic disease cohort from Cleveland Clinic.

In this study, we compared these six models in a cohort of adult spine patients to assess their ability to map PROMIS-GHS to EQ-5D in the spinal population.

## 2. Experimental Section

### 2.1. Surveys

A short demographics form was used to obtain gender, age, race/ethnicity, education, medical history, and spine diagnosis of participants.

The PROMIS Global Health survey includes ten global health items to assess overall health: (1) general health, (2) quality of life, (3) physical health, (4) mental health, (5) social satisfaction, (6) physical activities, (7) pain, (8) fatigue, (9) social activities, and (10) emotional distress. Every item except the pain item is rated on a numeric five-level scale (1 representing poor and 5 representing excellent); the pain item is scored from 0 to 10, where 0 indicates no pain and 10 indicates the worst imaginable pain. The pain item is then recoded to a five-level scale, and the fatigue and emotional problem item is recoded such that a high score represents better health status. Individual global item scores from completed PROMIS surveys were used to calculate estimates of EQ-5D index scores.

The EQ-5D is a preference-based instrument designed to measure generic health status across five dimensions of health: (1) mobility, (2) self-care, (3) usual activities, (4) pain/discomfort, and (5) anxiety/depression, with three response levels (no problems, some problems, extreme problems) [14]. A unique EQ-5D health state is defined by combining one level from each of the five dimensions, and each health state corresponds to a health index ranging from −0.109 to 1.0, with greater scores correlating to better overall health [15]. This index was calculated for every completed EQ-5D survey according to the valuations developed by Shaw et al. and derived from a large scale survey of the US general population [15]. The single visual analogue scale component of EQ-5D (EQ-5D VAS) was obtained but not evaluated in this study. Permission to use EQ-5D was granted by the EuroQol Group.

### 2.2. Study Design and Participants

This study was primarily conducted in the adult spine clinics of the three neurosurgeons (K.A., J.S.C, and L.K) at Yale University School of Medicine in New Haven, CT, with Institutional Review Board approval. Figure 1 illustrates the design of the study. In these clinics, 146 adult (>18 years of age) spine patients were recruited in 2017 as they entered the clinic with voluntary consent regardless of their clinical status (pre-operative, post-operative, or non-operative). Three forms were administered in paper to these patients: a demographics short form, PROMIS-GHS, and EQ-5D. PROMIS-GHS and EQ-5D were administered in random order. Completion of these two survey components was essential for obtaining an EQ-5D index and corresponding index estimates from PROMIS Global Health items. Out of 146 patients, complete survey responses were obtained from 121 patients.

### 2.3. Models Tested in the Study

**REV_Reg_:** This model was developed in 2009 by applying ordinary least squares (OLS) regression on the PROMIS Wave 1 Sample (i.e., the sample used by Revicki et al.) [13,16] to predict EQ-5D index scores from PROMIS-GHS items. This model uses eight out of 10 PROMIS-GHS items in its algorithm (excluding responses to general health and social satisfaction) and treats these items as continuous variables.

**REV_LE_**: This model is the result of applying linear equating, a method of linking, to REV_Reg_. While regression models aim to predict preference-based scores from profile-based responses, linking models align score distributions of observed and predicted scores to establish a scale that provides an equivalent preference-based score for each set of profile-based responses. Linear equating is applied to REV_Reg_ with the following equation:(1)$$Y_{LE}=\;\mu_{Y}+\frac{\sigma_{Y}}{\sigma_{Y_{R}}}\left(Y_{R}-\mu_{Y_{R}}\right)$$
where $Y_{LE}$ is the estimated value from linear equating, $\mu_{Y}\;$ and $\sigma_{Y}$ are the mean and standard deviation of the observed EQ-5D scores from the PROMIS Wave 1 Sample, respectively, and $\mu_{Y_{R}}$ and $\sigma_{Y_{R}}$ are the mean and standard deviation of the predicted EQ-5D scores from REV_Reg_, respectively.

**REV_equip_**: This model was developed by applying equipercentile equating to REV_Reg_. Equipercentile equating is a linking method that matches the cumulative distribution functions of observed scores and predicted scores from REV_Reg_ using smoothing functions or nonparametric techniques.

**CAT_Reg_**: This model was implemented in 2017 by Thompsons et al. Like REV_Reg_, this model utilizes OLS regression on the PROMIS Wave 1 sample to predict EQ-5D index scores from eight PROMIS-GHS items. Unlike REV_Reg_, CAT_Reg_ treats these items as categorical variables.

**CAT_LE_**: This model is the result of applying linear equating to CAT_Reg_.

**CAT_equip_**: This model was developed by applying equipercentile equating to CAT_Reg_.

### 2.4. Statistical Analysis

Statistical analyses were conducted in R Studio [17]. Responses to each of the 121 completed EQ-5D surveys were utilized to calculate an EQ-5D index score according to the valuations developed by Shaw et al. [15]. Estimates of the EQ-5D index scores from PROMIS Global Health Item responses were obtained by applying the six models developed by Revicki et al. and Thompson et al. (REV_Reg_, REV_LE_, REV_equip_, CAT_Reg_, CAT_LE_, CAT_equip_) [9,13].

The goodness of fit for each model in our sample of patients was measured with the Pearson correlation coefficient (*r*), coefficient of Determination (*R*^2^), mean squared error (MSE), and mean absolute error (MAE). Correlation r measures the strength of the linear relationship. Higher absolute values indicate stronger linear correlations. *R*^2^ demonstrates how much variance could be explained by the regression model. The mean squared error (MSE) and mean absolute error (MAE) were measured to examine the scale of difference between each estimate and observed value. Models with lower MSE or MAE have better predictions.

In addition, comparisons of actual EQ-5D scores and optimal estimates were performed by subgroups, such as gender, age groups, ethnicity, education, and spine diagnosis. According to Luo et al., 0.04 was recommended as the minimal clinically important difference of a EQ-5D utility score with a scale from −0.109 to 1 [18]. If the mean difference is less than 0.04, we consider it is an accurate estimate of the mean.

However, good linear correlation does not always imply good agreement. In order to evaluate the transformation on an individual level, the Bland–Altman assessment of agreement was conducted. It could visually show the difference between actual and estimated scores of each patient. Histograms of the observed EQ-5D scores and estimates from each model were also plotted to show distributions of scores.

## 3. Results

### 3.1. Demographic Characteristics

Table 1 contains the demographics of the experimental cohort of adult spine patients. Our cohort of 121 patients had an average age of 59 years, was 59% female, and had a majority with Caucasian race/ethnicity. Highest level of education in these patients ranged from less than high school (4%) to advanced college degree (17%), with 33% completing high school, 31% having some college or associate’s degree, and 14% having a bachelor’s degree. Patients had a variety of conditions in their medical histories, including cancer, lung disease, psychiatric illness, heart disease, rheumatologic disease, central nervous system (CNS) disorders, and liver/kidney disease.

The cohort of this study had demographics comparable to the sample of the generic US population studied by Revicki et al. [9] and the neurologic disease cohort studied by Thompson et al. [13]. Unlike Revicki et al. and Thompson et al., however, all sample subjects had spine diagnoses, including cervical and lumbar stenosis (most common), deformity, myelopathy, radiculopathy, spondylolisthesis, fracture, tumor, and pseudoarthrosis. The specificity of spine diagnosis distinguishes the cohort of this study from the general cohort of Revicki’s study.

### 3.2. Statistical Analysis

Table 2 presents the metrics used to assess the models applied to our sample. The estimated score in the CAT_Reg_ model (0.60) was closest to the observed EQ-5D index scores (0.62). The mean difference was 0.012 (95% CI, –0.012–0.036, *p* = 0.3144), which indicated no significant difference between actual EQ-5D score and CAT_Reg_ estimates. All other estimates were significantly different using the paired t-test. The *R*^2^ values for all six models ranged between 0.54 and 0.59. Pearson correlation coefficients were all above 0.7, showing strong linear correlation. Of the six models, CAT_Reg_ had the highest *R*^2^ (0.59) and lowest MSE (0.02) and MAE (0.11). Thus, CAT_Reg_ is the optimal model among them.

In order to investigate the accuracy of CAT_reg_ model predictions, subgroup analysis was also performed, shown in Table 3. Within most subgroups, the mean difference was less than 0.04 (the minimal clinically important difference of EQ-5D score), which means the EQ-5D score could be accurately predicted using PROMIS-GHS. For example, the female spine patients’ observed EQ-5D score was 0.62 and the estimate of CATreg was 0.60 (95% CI, 0.56–0.64), while the males’ was 0.60 vs. 0.60 (95% CI, 0.55–0.66). Caucasian Americans had a higher average EQ-5D score (actual 0.64 vs. estimates 0.64) than other ethnicities (actual 0.52 vs. estimates 0.50). The actual score for different education level ranged from 0.53 to 0.70. Generally, the larger the group size, the better prediction was achieved. All the subgroups with more than 17 patients had a mean difference less than 0.04, which indicates this score transformation should be more appropriately used on a group level, instead of individual level.

In order to investigate the prediction performance at an individual level, Bland-Altman analysis was conducted. Figure 2 demonstrated the mean residual was 0.01, with 95% limits of agreement between actual and CAT_reg_ estimated EQ-5D scores ranged from −0.25 to 0.27. It revealed that for a single patient, the variation from their actual score is huge and largely exceeded the minimal clinically important difference 0.04.

Figure 3 depicts histograms of the observed EQ-5D-3L scores and estimates from REVReg, REVLE, and REVequip in our sample. Figure 4 depicts histograms of the observed EQ-5D scores and estimates from CAT_Reg_, CAT_LE_, and CAT_equip_. In both figures, the histograms of regression estimates and linear equating estimates resemble a normal distribution, while the histograms of the observed 3L scores and equipercentile equating estimates have a bimodal distribution. These histograms confirmed that the estimates from the transformation models are not a good match on an individual level.

## 4. Discussion

### 4.1. Validation and Technical Aspects

Our study assessed and compared six models that were developed in a generic sample to map PROMIS-GHS to EQ-5D in a specific sample of patients with spinal disorders. In our sample of patients with spinal disease, all six models achieved an *R*^2^ greater than 0.5. According to Brazier et al., models that map to preference-based scores commonly achieve an *R*^2^ of greater than 0.5 within the sample of model development [19]. *R*^2^ as a measure of goodness-of-fit can determine how well the model explains the dataset it was estimated on. However, it did not show the scale of difference. In that regard, MSE and MAE can better assess mapping functions by indicating size of prediction errors [19]. So, we compared the models with consideration of all the goodness-of-fit indicators.

First, we agreed that treating PROMIS-GHS item scores 1 to 5 as categorical variables (CAT_Reg_) performed better than treating them as continues variables (REV_Reg_), with closer mean estimate (0.60 vs. 0.57, actual score = 0.62), higher *R*^2^ (0.59 vs. 0.57), and lower MAE (0.11 vs. 0.13). However, unlike the recommendation of using equating technics used in the Thompson et al. article, in our spine sample population, the linear and equipercentile equating models (REV_LE_, and REV_equip_, CAT_LE_, and CAT_equip_) did not work well compared to the CAT_Reg._ Thus, although all six models demonstrated adequate prediction ability, the CAT_Reg_ model is the optimal one for patients with spine disease.

Second, we recommend using this transformation from PROMIS-GHS to EQ-5D utility score on group-level mean estimates, not for individual prediction. From the subgroup analysis, it showed the accurate prediction (mean difference less than 0.04) was achieved in groups with more than 17 patients. To be more conservative, sample sizes of at least 30 patients are suggested for the good mean estimate of a EQ-5D score from PROMIS-GHS using the CAT_Reg_ model.

### 4.2. Utility of Health Care Measurement

HSMs have often been validated in patients with spinal disease before clinical application. For instance, Guilfoyle et al. validated the Medical Outcomes Study Short Form (SF-6, -12, -36), a general health outcome measure, in patients with lumbar disc prolapse, lumbar canal stenosis, and degenerative cervical myeloradiculopathy. This study found strong correlation between SF surveys and disease-specific measures such as the Roland Morris Disability Score (RMDS), Myelopathy Disability Index (MDI), and Hospital Anxiety and Depression Scales (HADS) [20,21]. Similarly, EQ-5D was assessed for its validity for use in spine surgery by comparison with the Oswestry Disability Index (ODI) in a study of patients who underwent lumbar spine surgery for degenerative disorders [21,22]. According to the study, EQ-5D and ODI were equal in assessment of health state, thus validating the use of EQ-5D in patients with spinal disorders.

The validation of EQ-5D and other HSM questionnaires in patients with spinal disorders paved the way for assessing the value of spinal interventions using health utility index scores. For instance, Witiw. et al. assessed the lifetime incremental cost-utility of surgical treatment for degenerative cervical myelopathy in a prospective observational cohort study by calculating health utility and QALYs from SF-6D [23]. Tosteson et al. used data from the Spine Patient Outcomes Research Trial (SPORT) to determine that lumbar discectomy was a clinically beneficial and cost-effective treatment of intervertebral disc herniation [24]. They also determined that spinal stenosis surgery was cost-effective but degenerative spondylolisthesis surgery was not cost-effective over a period of two years [25]. Conclusions from Tosteson et al. were based on the use of the EQ-5D index to obtain measures of QALY and incremental cost-effectiveness ratio.

Cost-utility studies of spinal interventions have also used estimation models to obtain health utility scores from other surveys. Qureshi et al. investigated the cost-effectiveness of anterior cervical discectomy and fusion (ACDF) and cervical disc replacement (CDR) as therapies for single-level cervical degenerative disc disease (DDD) [26]. To do this, the group used results of the 36-Item Short Form Health Survey (SF-36) from the ProDisc-C investigational device exemption study along with a model generated to estimate preference-based index scores from the Short Form-6 dimensions (SF-6D) (derived from a subsection of SF-36 items) [27].

Mapping PROMIS to EQ-5D can prove to be a powerful method of calculating health utility in economic cost-benefit studies. Along with its increased use by the NIH, PROMIS and its domain item banks allow flexibility in administration using either targeted short forms or computerized adaptive tests [4,9]. The importance of validating models such as those developed by Revicki et al. [9] and Thompson et al. [13] lies in assessing the clinical and economic utility of applying generic models to disease-specific populations, including those with spinal pathologies.

### 4.3. Clinical Implications

The findings in the paper indicate that PROMIS can act as a reasonable surrogate for EQ-5D. For hospitals or medical centers that have already collected PROMIS-GHS and do not have EQ-5D, they could use this transformation to estimate EQ-5D scores and then calculate the quality adjusted life-year for cost-effectiveness analysis. Based on previous reports and our data, it appears that CAT_Reg_ is the choice with the lowest error for patients with spinal disorders.

Measurement of health status not only assesses general cost-effectiveness of interventions but also provides the opportunity to assess individual patients longitudinally. Consequently, one can assess changes in conservative management, and treatment modalities can be altered in accordance to health status. Regular clinical usage of HSMs develops a general repository of health outcomes data that would otherwise come solely from research studies, potentially alleviating substantial costs for prospective research studies.

### 4.4. Limitations

One of the limitations of this study was the sample size of the cohort. Though the cohort in this study had a variety of spine pathologies, our sample size was limited to clinics in a single institution. The results may not represent the whole spine population. Second, we tried to create our own prediction model. However, only three out of 10 PROMIS-GHS items (general health, social satisfaction, pain) were significant predictors due to the limited sample size.

## 5. Conclusions

This study assesses and compares six models that map PROMIS-GHS to EQ-5D index values in a population of patients with spinal disorders. All six models demonstrate adequate and comparable predictive performance in our sample, thus validating their economic utility. Among the six models, the CATreg model is recommended for spine patients. That is, EQ-5D utility scores could be most accurately estimated by the linear combination of eight significantly correlated items from PROMIS-GHS, while scores 1 to 5 for each item is treated as a categorical variable. In addition, we suggest using this transformation model for group-based estimates, instead of for individual patient’s EQ-5D score estimates. Validation studies of HSMs can lead to their application in cost-utility analyses.

## Figures and Tables

**Figure 1 jcm-08-01506-f001:**
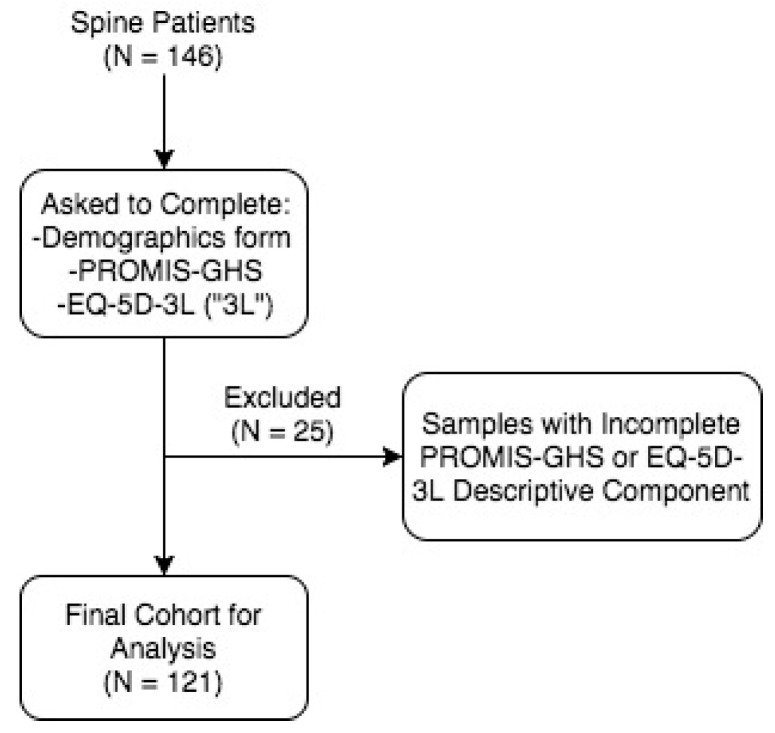
Sample Selection. PROMIS-GHS: Patient-Reported Outcomes Measurement Information System Global Health Survey; 5Q-5D: EuroQol 5-Dimension; 5Q-5D-3L; EQ-5D three-level survey.

**Figure 2 jcm-08-01506-f002:**
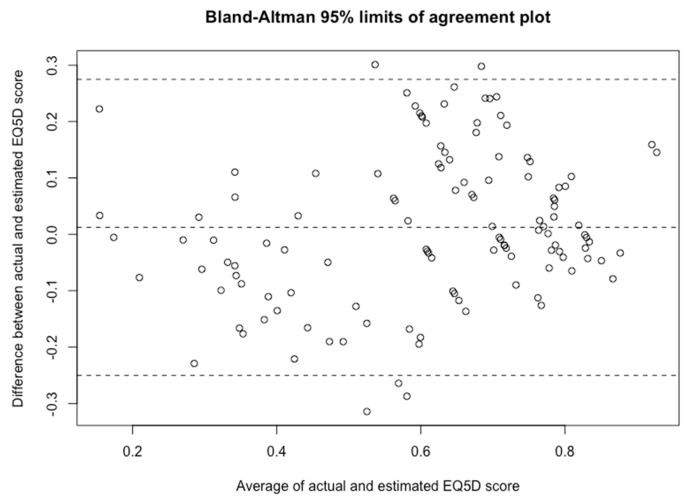
Bland-Altman agreement plot. *X* axis is the average score of the actual and estimated EQ-5D score of the CAT_reg_ model. *Y* axis is the difference between the two. Each dot represents a patient. The three dashed lines are upper 95% limits of agreements (mean + 1.96 SD), mean difference, and lower 95% limits of agreements (mean − 1.96 SD).

**Figure 3 jcm-08-01506-f003:**
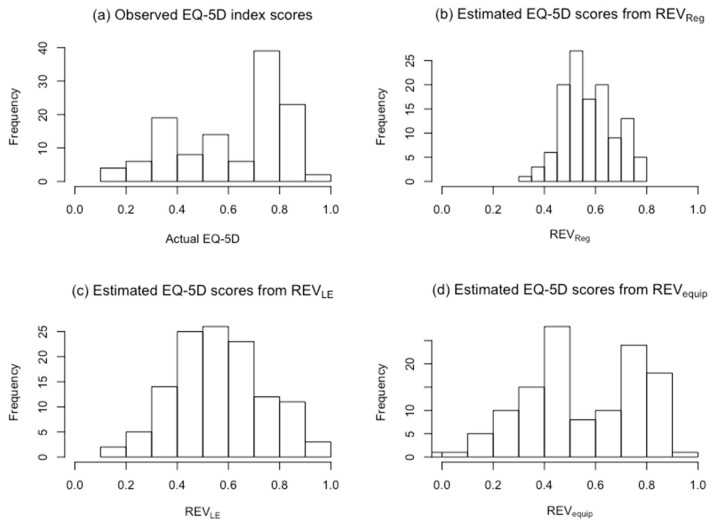
Histograms of observed EQ-5D index scores and estimates from REV_Reg_, REV_LE_, and REV_equip._

**Figure 4 jcm-08-01506-f004:**
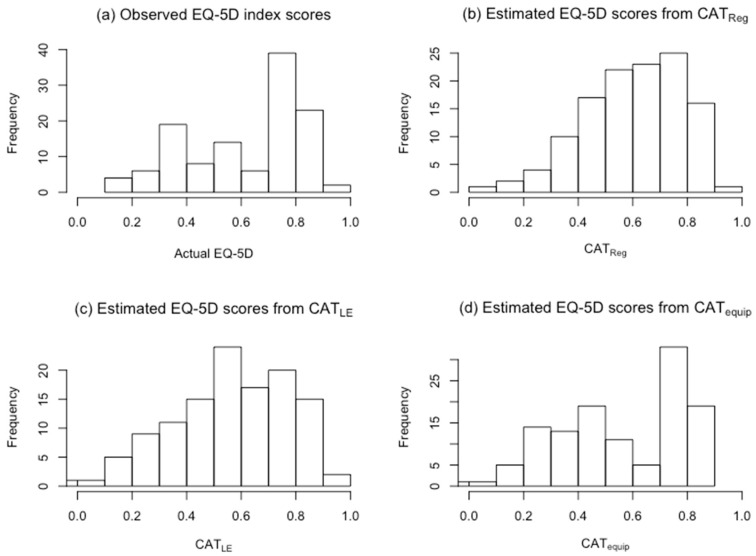
Histograms of observed EQ-5D index scores and estimates from CAT_Reg_, CAT_LE_, and CAT_equip._

**Table 1 jcm-08-01506-t001:** Demographic and clinical characteristics of survey participants.

Characteristic	Spine Patients (*N* = 121)
Age, mean ± SD	59 ± 13
Gender, *n* (%)	
Female	71 (59)
Male	49 (40)
Race/Ethnicity, *n* (%)	
Caucasian American	94 (77)
African American	13 (11)
Hispanic American	9 (7)
Caucasian American and Hispanic American	1 (1)
Asian American	1 (1)
Caucasian American and Native American	1 (1)
Highest Level of Education, *n* (%)	
Advanced Degree	21 (17)
Bachelor’s Degree	17 (14)
Some College or Associate’s Degree	38 (31)
High School Completion	40 (33)
Less than High School	5 (4)
Medical History, *n* (%)	
Psychiatric Illness	33 (27)
Lung Disease	30 (25)
Heart Disease	27 (22)
Cancer/Tumor	25 (21)
CNS disorders	18 (15)
Rheumatologic Disease	17 (14)
Liver/Kidney Disease	11 (9)
Spine Diagnosis, *n* (%)	
Stenosis	35 (29)
Radiculopathy	14 (12)
Myelopathy	13 (11)
Deformity	12 (10)
Disc Herniation	5 (4)
Spondylolisthesis	5 (4)
Fracture	3 (2)
Tumor	3 (2)
Pseudoarthrosis	1 (1)

**Table 2 jcm-08-01506-t002:** Mean (standard deviation (SD)) of actual and estimated EQ-5D Index Scores, *R*^2^ values, correlation coefficients, mean squared errors (MSE), and mean absolute errors (MAE) for models in the spine patient sample (*N* = 121).

	Mean (SD)	*R* ^2^	*r*	MSE	MAE
Actual	0.62 (0.21)				
REV_Reg_	0.57 (0.10)	0.57	0.76	0.02	0.13
REV_LE_	0.56 (0.17)	0.57	0.76	0.02	0.12
REV_equip_	0.54 (0.22)	0.57	0.76	0.03	0.12
CAT_Reg_	0.60 (0.18)	0.59	0.77	0.02	0.11
CAT_LE_	0.56 (0.22)	0.59	0.77	0.02	0.12
CAT_equip_	0.56 (0.23)	0.54	0.73	0.03	0.13

**Table 3 jcm-08-01506-t003:** Comparison of actual EQ-5D scores and estimates of the CAT_reg_ model by subgroups.

	*N*	Actual EQ-5D Mean (SD)	CATreg Estimates Mean (SD)	Mean Difference
Gender				
Female	71	0.62 (0.20)	0.60 (0.16)	0.02
Male	49	0.60 (0.22)	0.60 (0.20)	0.00
Age groups, years				
18–45	17	0.59 (0.22)	0.54 (0.21)	0.05
46–65	63	0.60 (0.23)	0.59 (0.18)	0.01
65+	40	0.65 (0.16)	0.65 (0.15)	0.00
Ethnicity				
Caucasian American	94	0.64 (0.20)	0.64 (0.16)	0.00
Others	27	0.52 (0.23)	0.50 (0.18)	0.02
Highest education level				
Advanced degree	21	0.65 (0.16)	0.69 (0.18)	−0.04
Bachelor’s degree	17	0.70 (0.16)	0.62 (0.16)	0.08
Some college or associate’s degree	38	0.61 (0.22)	0.61 (0.15)	0.00
High school completion	40	0.58 (0.23)	0.56 (0.19)	0.02
Less than high school	5	0.53 (0.27)	0.48 (0.19)	0.05
Spine Diagnosis				
Stenosis	35	0.65 (0.20)	0.61 (0.17)	0.04
Other	22	0.59 (0.22)	0.58 (0.20)	0.01
Radiculopathy	14	0.65 (0.20)	0.63 (0.20)	0.03
Myelopathy	13	0.60 (0.24)	0.60 (0.21)	0.00
Deformity	12	0.62 (0.19)	0.61 (0.16)	0.01
Disc herniation	5	0.58 (0.27)	0.64 (0.25)	−0.06
Spondylolisthesis	5	0.50 (0.29)	0.48 (0.20)	0.02
Unknown	4	0.60 (0.18)	0.63 (0.14)	−0.03
Fracture	3	0.63 (0.06)	0.71 (0.02)	−0.08
Tumor	3	0.63 (0.06)	0.56 (0.07)	0.07
Herniated disc	2	0.31 (0.00)	0.50 (0.05)	−0.19
Pseudoarthrosis	1	0.44	0.72	−0.29

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
