# Peer review of "Validating the Transformation of PROMIS-GH to EQ-5D in Adult Spine Patients"

_jcm, 2019, doi:10.3390/jcm8101506_

Round 1

Reviewer 1 Report

This manuscript deals with the comparison of several PROMIS-GHS to EQ-5D-3L mapping methods in adult patients with spine disorders in order to determine the best fit for this population. This paper is well written and can be very useful in both clinical practice and medicoeconomic research. However, there are some minor points that need to be discussed.

If i understand correctly, you used the regression coefficients of the REVreg model developed in 2009 by Revicki et al to predict the EQ-5D-3L index scores using the the PROMIS-GHS scores as regressors. Have you tried developping a specific model focused on the spine population using your data? Maybe, a spine specific model could lead to a better fit for this population than the Revicki et al model (due to the heterogeneity of the Revicki et al population). And even with a small sample size, a correction of the predicted EQ-5D values using simple linear regression could be proposed.

The heteroscedasticity of the Revicki at al model could have also been studied by comparing the model errors between the different sub-populations (depending on spine diagnosis and medical history). Could you comment on that?

Reviewer 2 Report

This is a well-written paper and the topic is interesting. However,  there are several points to be addressed.

The authors should mention more about inclusion criteria. ASD is heterogeneous and what exactly was your inclusion criteria (thoracolumbar only or cervical deformity included? included untreated AIS? etc.) The authors should elaborate more about Fig 2 and 3. It is hard to understand why these figures are present. The conclusion is obscure. Which formula is the best? If every formula can be equally used, which one should we use? Please mention your conclusion based on your results. As you mention, every formula can be used. But, is it better to have patients fill in EQ5D, not converting from PROMIS?

Round 2

Reviewer 2 Report

Authors successfully corrected their manuscript. I am satisfied with that. I personally think this paper is suitable for publication.